# Assessment of Bidirectional Relationships between Leisure Sedentary Behaviors and Neuropsychiatric Disorders: A Two-Sample Mendelian Randomization Study

**DOI:** 10.3390/genes13060962

**Published:** 2022-05-27

**Authors:** Qian He, Adam N. Bennett, Beifang Fan, Xue Han, Jundong Liu, Kevin Chun Hei Wu, Ruixuan Huang, Juliana C. N. Chan, Kei Hang Katie Chan

**Affiliations:** 1Department of Biomedical Sciences, City University of Hong Kong, Hong Kong SAR 999077, China; qhe226-c@my.cityu.edu.hk (Q.H.); a.n.bennett@my.cityu.edu.hk (A.N.B.); jdliu4-c@my.cityu.edu.hk (J.L.); chunhwu@cityu.edu.hk (K.C.H.W.); 2Department of Mental Health, Shenzhen Nanshan Center for Chronic Disease Control, Shenzhen 518000, China; fanbf@outlook.com (B.F.); xuehan_sz@hotmail.com (X.H.); 3Department of Electrical Engineering, City University of Hong Kong, Hong Kong SAR 999077, China; rxhuang4-c@my.cityu.edu.hk; 4Li Ka Shing Institute of Health Sciences, The Chinese University of Hong Kong, Hong Kong SAR 999077, China; jchan@cuhk.edu.hk; 5Hong Kong Institute of Diabetes and Obesity, The Chinese University of Hong Kong, Hong Kong SAR 999077, China; 6Department of Medicine and Therapeutics, The Chinese University of Hong Kong, Hong Kong SAR 999077, China; 7Department of Epidemiology, Centre for Global Cardiometabolic Health, Brown University, Providence, RI 02912, USA

**Keywords:** schizophrenia, Alzheimer’s disease, major depressive disorder, sedentary behaviors, Mendelian randomization

## Abstract

(1) Background: Increasing evidence shows that sedentary behaviors are associated with neuropsychiatric disorders (NPDs) and thus may be a modifiable factor to target for the prevention of NPDs. However, the direction and causality for the relationship remain unknown; sedentary behaviors could increase or decrease the risk of NPDs, and/or NPDs may increase or decrease engagement in sedentary behaviors. (2) Methods: This Mendelian randomization (MR) study with two samples included independent genetic variants related to sedentary behaviors (*n* = 408,815), Alzheimer’s disease (AD; *n* = 63,926), schizophrenia (SCZ; *n* = 105,318), and major depressive disorder (MDD; *n* = 500,199), which were extracted from several of the largest non-overlapping genome-wide association studies (GWASs), as instrumental variables. The summarized MR effect sizes from each instrumental variable were combined in an IVW (inverse-variance-weighted) approach, with various approaches (e.g., MR-Egger, weighted median, MR-pleiotropy residual sum and outlier), and sensitivity analyses were performed to identify and remove outliers and assess the horizontal pleiotropy. (3) Results: The MR evidence and linkage disequilibrium score regression revealed a consistent directional association between television watching and MDD (odds ratio (OR), 1.13 for MDD per one standard deviation (SD) increase in mean television watching time; 95% CI, 1.06–1.20; *p* = 6.80 × 10^−5^) and a consistent relationship between computer use and a decrease in the risk of AD (OR, 0.52 for AD per one SD increase in mean computer use time; 95% CI, 0.32–0.84; *p* = 8.20 × 10^−3^). In the reverse direction, MR showed a causal association between a reduced risk of SCZ and an increase in driving time (β, −0.016; 95% CI, −0.027–−0.004; *p* = 8.30 × 10^−3^). (4) Conclusions: Using genetic instrumental variables identified from large-scale GWASs, we found robust evidence for a causal relationship between long computer use time and a reduced risk of AD, and for a causal relationship between long television watching time and an increased risk of MDD. In reverse analyses, we found that SCZ was causally associated with reduced driving time. These findings fit in with our observations and prior knowledge as well as emphasizing the importance of distinguishing between different domains of sedentary behaviors in epidemiologic studies of NPDs.

## 1. Introduction

Neuropsychiatric disorders (NPDs), such as Alzheimer’s disease (AD), schizophrenia (SCZ), and major depressive disorder (MDD), are characterized by changes in cognition, mood, and/or behavior and are caused by altered neuronal pathology or abnormal physiological conditions [1]. Cancer and cardiovascular disease are the leading cause of death; however, the morbidity and quality of life lost may be greater amongst those with NPDs [2]. As reported, NPDs comprise more than 10% of disabilities worldwide. Despite the prevalence of NPDs, few risk factors have been established, and identifying modifiable factors to target for prevention has been especially challenging [3]. One promising target for modification is sedentary behaviors, which are known as waking behaviors with an energy expenditure of less than one and a half hour’s metabolic equivalents that are performed in sitting, reclining, or lying postures [4]. Observational studies have indicated that the more time engaged in sedentary behaviors, the higher the risk of NPDs [5,6,7]. Several meta-analyses of randomized clinical trial data have suggested that sedentary behaviors may increase the risk of AD [8,9], SCZ [5,10], and MDD [7,11], and prospective studies have indicated that the higher the physical activity levels, the lower the NPD incidence [12,13,14].

However, although sedentary behaviors have been assessed as a potential risk factor for NPDs, several questions remain to be answered. First, do sedentary behaviors causally influence NPD risk or vice versa. Several studies have reported that patients with SCZ and MDD had a lower level of physical activity and were more likely to live a sedentary lifestyle [5,15]; thus, the sedentary behavior–NPD relationship may be explained by reverse causation. Second, are there differences in the effects on the risk of NPDs between the effects of various domains of sedentary behavior, such as mentally passive and active sedentary behaviors? Although no clear definitions of mentally passive and mentally active sedentary behaviors have been established, television watching, sitting, and listening are generally considered ‘mentally passive’ behaviors, whereas using a computer, reading books or newspapers, knitting, attending a meeting, and car driving are generally considered ‘mentally active’ behaviors [7,11]. Inconsistent results were found on the relationship of mentally passive sedentary behaviors and mentally active sedentary behaviors with NPDs [11,16]. Third, is the association between sedentary behaviors and NPDs consistent when potentially confounding factors are minimized or adjusted? Although the design of randomized clinical trials reduces the influence of confounding factors, such trials are intensive to carry out, and their sample sizes have been relatively not large enough [17]. Moreover, it is difficult to use observational trials to exclude confounders, such as genetic, social, and behavioral factors [18]. Thus, it is still not clear whether there is a causal relationship between sedentary behaviors and NPDs, and high-quality evidence is needed to clarify the association.

Recently, GWASs have been performed to identify significant loci for sedentary behaviors [19] and NPDs [20,21,22]. In addition, Mendelian randomization (MR) designs make use of valid instrumental variables to enable robust evaluations of causal inference [7,19,23]. Genetic variants are such an instrument, as they are allocated randomly before birth and are established well before disease onset, which means that using genetic variants could avoid the influences of environmental factors. This minimizes the effects of residual confounding variables and gets rid of the reverse causation effects that typically limit observational studies. Therefore, in this study, we implemented a bidirectional MR design to evaluate whether a causal association exists between sedentary behaviors and NPD risk, and vice versa.

## 2. Methods and Materials

### 2.1. Data Sources and Instruments

We obtained the GWAS summary statistics based on the hg19 coordinate. Data source details are provided in Appendix A.

#### 2.1.1. Leisure Sedentary Behaviors

We focused on the summary statistics from a recent GWAS of the sedentary behaviors of participants in the UK Biobank Study [19]. This GWAS evaluated three continuous-leisure sedentary behavior phenotypes: (1) self-reported hours spent watching television per day; (2) self-reported hours spent using a computer (excluding work use) per day; and (3) self-reported hours spent driving per day. It identified 152 independent and significant genome-wide single-nucleotide polymorphisms (SNPs) for television watching (*n* = 408,815), 37 for computer use (*n* = 408,815), and 4 for driving (*n* = 408,815).

#### 2.1.2. AD

Summary statistics of AD were obtained from the most recent GWAS of clinical patients diagnosed with late-onset AD performed by the International Genomics of Alzheimer’s Project consortium [20]. This GWAS consisted of 21,982 cases and 41,944 controls (*n* = 63,926) and identified four independent and significant genome-wide SNPs for AD [20].

#### 2.1.3. SCZ

SCZ summary statistics were obtained from a recent GWAS conducted by the Psychiatric Genomics Consortium (PGC) [22]. This GWAS consisted of 40,675 cases and 64,643 controls (*n* = 105,318) and identified 81 independent and significant genome-wide SNPs for SCZ.

#### 2.1.4. MDD

We focused on the GWAS summary statistics from the most recent and largest studies of MDD [21]. The dataset consisted of 170,756 cases and 329,443 controls (*n* = 500,199) and identified six independent and significant genome-wide SNPs for MDD.

### 2.2. Linkage Disequilibrium Score Regression (LDSC) of Genetic Correlation

We assessed the genetic correlation (*rg*) between sedentary behaviors and NPDs by applying LDSC [24]. GWAS summary statistics were filtered based on the HapMap3 protocols [25]. Genetic variants were not included in the further analyses if they had an ambiguous strand, had a frequency of minor alleles of less than 0.01, or were located in the major histocompatibility complex region (chromosome 6: 28,477,797–33,448,354) because of the complex linkage disequilibrium (LD) structure in the location [26].

### 2.3. Instrument Variable Selection

Selected genetic variants should be examined to assess whether they satisfy the three MR assumptions (Figure 1). Briefly, to be regarded as correlating sedentary behaviors and NPDs, genetic variants need to (1) be robustly associated with sedentary behaviors, (2) be unrelated to factors confounding the association between exposure and outcome, and (3) have effects on the risk of NPDs only via their effects on sedentary behaviors. Accordingly, we selected the following set of genetic instruments: (1) genetic variants meeting a strict *p* threshold (*p*  <  1 × 10^−8^); (2) clumped SNPs according to *r*^2^, for the independence of SNPs (i.e., only one representative SNP was retained when SNPs were correlated at *r*^2^ > 0.01), restricted to the European ancestry reference data from the 1000 Genomes Project; (3) overlapped proxy SNPs with a high LD (*r*^2^ > 0.80), identified by an LDproxy online search based on LDlink (https://ldlink.nci.nih.gov/, accessed on 10 October 2021), as a replacement for any SNPs for the exposure trait that were not available in the GWAS summary statistics of the outcome trait; and (4) only European-ancestry participants in the MR study, with population stratification excluded to prevent violation of the independence assumption. In addition, we searched each instrumental SNP and its proxies in the PhenoScanner GWAS database (v2; http://www.phenoscanner.medschl.cam.ac.uk/, accessed on 10 October 2021)) and GWAS catalogue (https://www.ebi.ac.uk/gwas/, accessed on 10 October 2021)) to identify any previous associations with possible confounding phenotypes, and evaluated the influences of excluding these genetic variants from the two-sample MR analysis manually.

### 2.4. Statistical Analyses

MR analyses were performed in R using the ‘TwoSampleMR (v 0.4.20)’ and MR-PRESSO (MR Pleiotropy Residual Sum and Outlier; version 1.0) [27] packages. These packages harmonize the GWAS summary statistics of exposure and outcome including information such as SNP IDs, effect and reference alleles, effect sizes (odds ratios (ORs) should be converted to β by the transformation of log), standard errors (SEs), *p*-values, and frequencies of effect alleles for genetic variants. Effect estimates are reported as β statistics for continuous outcomes (i.e., self-reported leisure sedentary behavior hours) and as ORs for dichotomous outcomes (i.e., AD, SCZ, and MDD status). In all two-sample MR analyses, a two-sided *p*-value of less than 0.05 was viewed as statistically significant.

### 2.5. Pleiotropy in MR Analyses

In analyses of two-sample MR, pleiotropy represents those genetic variants or SNPs with multiple effects. That is, pleiotropic genetic variants or SNPs may have an effect on the outcome, not the exposure, which could cause a bias in the MR estimate and potential confounding effects; thus, investigating pleiotropy is essential [19]. For every two directions of potential estimate assessment, we adopted the inverse-variance-weighted (IVW) meta-analysis approach to summarize the MR estimates. The IVW approach is viewed as a weighted regression for the genetic variant–outcome influence on the genetic variant–exposure influence with the intercept constrained to zero. The I^2^ index [28] and Cochran’s Q [29] statistics were used to assess the heterogeneity generated by the combination of different genetic variants or SNPs in the fixed effects IVW approach. Heterogeneity statistics could help in providing extra information on pleiotropy because low heterogeneity denotes that estimates between genetic variants or SNPs are different only by accident, which is only due to the absence of pleiotropic effects. An I^2^ index value higher than 25% and a *p*-value of Cochran’s Q test of less than 0.05 indicate the presence of moderate to high heterogeneity and, consequently, pleiotropy [29]. In cases where Cochran’s Q test indicated there was pleiotropy, we adopted the results of a random effects model instead of a fixed effects IVW model [29].

Next, an MR-Egger test was conducted. The intercept was freely included and estimated in the MR-Egger regression to evaluate the average pleiotropic bias between genetic variants. Moreover, we assessed the heterogeneity within the MR-Egger analysis by calculating the statistic of Rucker’s Q [29]. A significant difference (*p* < 0.05) between the statistics of Cochran’s Q and Rucker’s Q (Q−Q’) [29] denotes that the Egger test is a more suitable approach to assess the genetic relationship between a certain exposure and outcome.

We also used MR-PRESSO to detect any outliers in all reported results that likely reflected pleiotropic biases [30], and to correct for these. We did not include those genetic variants or SNPs with potential pleiotropic effects across all phenotypes, with these variants identified by querying the PhenoScanner database and GWAS catalogue for SNPs with an LD higher than 0.8. We adopted the MR Steiger filtering approach in the primary analysis to exclude genetic variants more closely associated with exposure (NPDs) than outcome (sedentary behaviors) [31]. We also used MR Steiger filtering to calculate the coefficient of determination (R^2^) for the exposure and outcome. Variants were removed if the exposure R^2^ was significantly lower than the outcome R^2^ [31].

Additionally, we used other established MR methods that generate estimates that are relatively robust to horizontal pleiotropy but have lower statistical power than the methods described above. We also used the weighted median approach [32], which treats the median MR estimate as the estimate for the assessment of a causal association, and the weighted mode approach [33], which allows most of the genetic variants to be invalid in case the most significant numbers that generate similar MR estimates are valid.

### 2.6. Weak Instrumental Bias in MR Analyses

The strength of genetic instrumental variables was evaluated by the *F* value. This was calculated by the equation *F* = R^2^(*n* − 2)/(1 − R^2^) [34], where R^2^ is the proportion of variances of sedentary behavior explained by the genetic variances or SNPs, and *n* is the sample size [34]. An *F* value higher than 10 means a possible low risk of weak instrumental bias in MR analyses [34] which is important to prevent violation of the ‘NO Measurement Error’ assumption.

## 3. Results

### 3.1. Genetic Correlation between Sedentary Behaviors and NPDs

We performed bivariate LDSC with and without a constrained intercept and identified strong shared genetic correlations between sedentary behaviors and NPDs (Table 1). Significantly positive genetic correlations between AD and television watching were detected (constrained intercept *rg* = 0.15, standard error (SE) = 0.026, *p* = 6.74 × 10^−9^; unconstrained intercept *rg* = 0.22, SE = 0.07, *p* = 1.50 × 10^−3^). Significantly negative genetic correlations between AD and computer use were detected (constrained intercept *rg* = −0.17, SE = 0.026, *p* = 1.59 × 10^−11^; unconstrained intercept *rg* = −0.25, SE = 0.068, *p* = 2.00 × 10^−4^). The genetic correlation between AD and driving showed a positive association with a constrained intercept and a non-significant association with an unconstrained intercept (constrained intercept *rg* = 0.093, SE = 0.034, *p* = 6.08 × 10^−3^; unconstrained intercept *rg* = 0.11, SE = 0.082, *p* = 0.172).

Significantly negative genetic correlations were found between SCZ and television watching (constrained intercept *rg* = −0.08, SE = 0.011, *p* = 1.97 × 10^−11^; unconstrained intercept *rg* = −0.098, SE = 0.018, *p* = 2.02 × 10^−8^), computer use (constrained intercept *rg* = −0.075, SE = 0.014, *p* = 3.28 × 10^−8^; unconstrained intercept *rg* = −0.068, SE = 0.021, *p* = 1.12 × 10^−3^), and driving (constrained intercept *rg* = −0.16, SE = 0.017, *p* = 3.04 × 10^−20^; unconstrained intercept *rg* = −0.19, SE = 0.27, *p* = 5.93 × 10^−12^).

Significantly positive genetic correlations were detected between MDD and television watching (constrained intercept *rg* = 0.19, SE = 0.013, *p* = 9.87 × 10^−52^; unconstrained intercept *rg* = 0.13, SE = 0.019, *p* = 9.24 × 10^−11^). The positive genetic correlation between MDD and computer use was significant with a constrained intercept and non-significant with an unconstrained intercept (constrained intercept *rg* = 0.039, SE = 0.017, *p* = 0.0203; unconstrained intercept *rg* = 0.018, SE = 0.026, *p* = 0.477). Significantly negative genetic correlations were detected between MDD and driving (constrained intercept *rg* = −0.16, SE = 0.02, *p* = 1.02 × 10^−17^; unconstrained intercept *rg* = −0.07, SE = 0.029, *p* = 0.0222).

### 3.2. MR Analyses

The lists of instrumental SNPs of sedentary behaviors (television watching, computer use, and driving) for AD, SCZ, and MDD obtained from the MR analyses are provided in Appendix A. The lists of instrumental SNPs of NPDs (AD, SCZ, and MDD) for sedentary behaviors are provided in Appendix A. The sedentary behavior instruments had *F* values ranging from 37 to 41, and the NPD instruments had *F* values ranging from 14 to 28, indicating there was a low chance of a weak instrumental bias. Genetic instrumental variables explained 0.04% to 1.70% of the variance or liability of each phenotype of exposure (Appendix A).

#### 3.2.1. Sedentary Behaviors and AD

We conducted MR analysis to explore the relationship between television watching (using the 85 most significant SNPs), computer use (using the 22 most significant SNPs), and driving (using the 4 most significant SNPs) and AD. The results of the MR-IVW random effects approach indicate that a causal effect was found between computer use and AD (OR, 0.52 for AD per one standard deviation (SD) increase in mean computer use time; 95% CI, 0.32–0.84; *p* = 0.0082), indicating computer use was related to a decreased risk of AD.

We then adopted reverse MR analyses to test the causal association between AD and sedentary behaviors (using the four most significant SNPs), and the result shows no evidence of causal relationships between AD and sedentary behaviors.

The results are shown in Figure 2 and Figure 3 (forest plots), and in Appendix A.

Using the MR-IVW fixed effects model, a causal association was found between computer use and AD (IVW OR, 0.52 for AD per one standard deviation (SD) increase in mean computer use time; 95% CI, 0.32–0.84; *p* = 0.0082), indicating computer use was related to a decreased risk of AD; the weighted median, weighted mode, and MR-Egger analysis yielded similar results (Appendix A). However, no causal effect was detected between television watching and AD (IVW OR, 1.16; 95% CI, 0.91–1.46; *p* = 0.219) or between driving and AD (IVW OR, 0.64; 95% CI, 0.22–1.92; *p* = 0.433). No variants were removed due to MR Steiger filtering of potential outliers detected by MR-PRESSO (Appendix A). Genetic variants that were excluded due to having potential pleiotropic effects across all phenotypes are summarized in Appendix A.

In order to identify pleiotropy, we estimated heterogeneity for MR-IVW analyses by using the I^2^ index and Cochran’s Q, and estimated heterogeneity for MR-Egger analyses by using Rucker’s Q (Appendix A). The I^2^ and Cochran’s Q indices indicated that there was heterogeneity in computer use and driving. We therefore adopted the IVW random effects model to assess the associations between AD and computer use (OR, 0.53; 95% CI, 0.30–0.92; *p* = 0.0024), television watching (OR, 1.15, 95% CI, 0.90–1.48; *p* = 0.244), and driving (OR, 0.65; 95% CI, 0.18–2.32, *p* = 0.504), which showed the same effects. The statistic of Rucker’s Q was not significantly lower than the statistic of Cochran’s Q for sedentary behaviors (Appendix A), showing no evidence of unbalanced horizontal pleiotropy as well as confirming the analyses of the IVW model as the best method. Scatter plots and forest plots were used to visualize the heterogeneity (Appendix A). Separate analyses with each SNP removed revealed that no single SNP drove the results (Appendix A, leave-one-out plot). Furthermore, a funnel plot is shown in Appendix A. The MR-Egger intercept *p*-values were higher than 0.05 (Appendix A), indicating an absence of bias due to pleiotropy in the MR-IVW analyses. In summary, there was an opposite causal relationship between computer use and AD.

In the other direction, an MR-IVW random effects approach was adopted to examine heterogeneity. Across all MR approaches, we failed to find a causal relationship between AD and television watching [IVW (random effects) β, 0.0022, 95% CI, −0.032–0.037, *p* = 0.901], computer use [IVW (random effects) β, 0.0061, 95% CI, −0.029–0.041, *p* = 0.734], and driving (IVW β −0.021, 95% CI, −0.057–0.014, *p* = 0.242; Figure 3 and Appendix A).

#### 3.2.2. Sedentary Behaviors and SCZ

We performed MR analyses to explore the association between television watching (using the top 98 SNPs), computer use (using the top 23 SNPs), and driving (using the top 4 SNPs) and SCZ. The results indicate that there was no causal relationship between sedentary behaviors and SCZ when adopting the appropriate approach.

We then performed reverse MR analyses to test the causal relationship between SCZ and sedentary behaviors (using the top 36 SNPs), and we found a causal relationship between SCZ and driving (β, −0.016; 95% CI, −0.027–−0.004; *p* = 0.0083), indicating SCZ was related to reduced driving time.

The results are shown in Figure 4 and Figure 5 (forest plots) and Appendix A.

Using the MR-IVW fixed effects model, a causal association was detected between computer use and SCZ (IVW OR, 0.62 for SCZ per one SD increase in mean computer use time; 95% CI, 0.47–0.82; *p* = 0.00074) and between driving and SCZ (IVW OR, 0.10 for SCZ per one SD increase in mean driving time; 95% CI, 0.05–0.19; *p* = 1.84 × 10^−11^). No causal effect was found between television watching and SCZ (IVW OR, 0.95 for SCZ per one SD increase in mean television watching time; 95% CI, 0.83–1.08, *p* = 0.449; Appendix A). The variants excluded in the MR-PRESSO analyses or removed due to the MR Steiger filtering approach are presented in Appendix A. The genetic variants excluded due to having potential pleiotropic effects in all phenotypes are shown in Appendix A.

Due to potential pleiotropy (Appendix A), we adopted the IVW random effects model to evaluate the relationship between television watching and SCZ [MR-IVW (random effects) OR, 0.95; 95% CI, 0.69–1.30, *p* = 0.754], which was unchanged. However, the relationship between computer use and SCZ was non-significant [MR-IVW (random effects) OR, 0.62; 95% CI, 0.30–1.28; *p* = 0.194] in this analysis. When we used the MR-Egger approach to assess the association between driving and SCZ, we found that the relationship was non-significant (OR, 0.00001; 95% CI, 0.00–125.03; *p* = 0.297; Appendix A).

The scatter plots and forest plots used to visualize heterogeneity are presented in Appendix A. Separate analyses with each SNP removed revealed that no single SNP drove the results (Appendix A, leave-one-out plot). The funnel plots of each SNP are shown in Appendix A. In summary, there was no causal relationship between sedentary behaviors and SCZ.

In the other direction, an MR-IVW random effects approach was adopted to address heterogeneity. We found no causal relationship between SCZ and television watching (β, −0.017; 95% CI, −0.037–0.0025; *p* = 0.0884) or between SCZ and computer use (β, −0.013; 95% CI, −0.028–0.0018; *p* = 0.0847). However, we did find a causal relationship between SCZ and driving (β, −0.016; 95% CI, −0.027–−0.004; *p* = 0.0082; Figure 6 and Appendix A). In summary, we found an opposite causal relationship between SCZ and driving, indicating SCZ was related to reduced driving time.

#### 3.2.3. Sedentary Behaviors and MDD

A series of MR analyses were conducted to assess the causal association between television watching (using the top 95 SNPs), computer use (using the top 22 SNPs), and driving (using the top 4 SNPs) and MDD. The MR-IVW random effects approach was adopted to assess the causal relationship between television watching and MDD (OR, 1.13; 95% CI, 1.02–1.25; *p* = 0.0229), indicating a long television watching time was related to a high risk of MDD.

We then performed reverse MR analyses to test the causal relationship between MDD and sedentary behaviors (using the top six SNPs), and the results show that no causal relationship was found between MDD and sedentary behaviors.

The results are shown in Figure 6 and Figure 7 and in Appendix A.

Using the MR-IVW fixed effects approach, a causal effect was detected between television watching and MDD (IVW OR, 1.13 for MDD per one SD increase in mean television watching time; 95% CI, 1.06–1.20; *p* = 6.80 × 10^−5^). No causal effect was found between computer use and MDD (IVW OR, 0.92 for MDD per one SD increase in mean computer use time; 95% CI, 0.81–1.085; *p* = 0.225) or between driving and MDD (IVW OR, 0.88 for MDD per one SD increase in mean driving time; 95% CI, 0.65–1.19; *p* = 0.404; Appendix A). All variants excluded in the MR-PRESSO analyses or removed due to MR Steiger filtering are presented in Appendix A. The genetic variants that were excluded due to having potential pleiotropic effects across all phenotypes are shown in Appendix A.

Due to heterogeneity, the IVW random effects model was adopted to evaluate the causal relationship between television watching and MDD (OR, 1.13; 95% CI, 1.02–1.25; *p* = 0.0229; Appendix A), which was unchanged. The MR-Egger analysis detected no causal relationship between computer use and MDD (OR, 3.19; 95% CI, 0.82–12.4; *p* = 0.10) or between driving and MDD (OR, 4.08; 95% CI, 0.12–7.02, *p* = 0.076; Appendix A). The scatter plots and forest plots used to visualize the heterogeneity are presented in Appendix A. Furthermore, separate analyses in which each SNP was removed revealed that no single SNP drove these results (Appendix A, leave-one-out plot). The funnel plots of each SNP are shown in Appendix A. In summary, there was a consistent causal relationship between television watching and MDD.

In the other direction, no causal relationship was detected between MDD and television watching [MR-IVW (random effects) β, −0.086; 95% CI, −0.19–0.02; *p* = 0.114], computer use (MR-Egger (heterogeneity) β, 0.305; 95% CI, −0.461–1.071; *p* = 0.179), or driving (MR-Egger (heterogeneity) β, −0.318; 95% CI, −0.259–0.065, *p* = 0.216; Figure 7 and Appendix A). In summary, no causal relationship was found between MDD and sedentary behaviors.

## 4. Discussion

NPDs are directly or indirectly related to cerebral dysfunction and contribute significantly to a high societal burden of mortality and morbidity in adult populations [35]. Researchers are therefore dedicated to developing disease-modifying treatments for NPDs. Thus, identifying effective strategies for preventing NPDs would assist in the improvement of global population health [36]. Recent evidence suggests that sedentary behaviors may be risk factors for NPDs [37]; however, observational studies seeking to establish relationships between sedentary behaviors and NPDs have yielded inconsistent results [6,16]. To determine causal relationships between sedentary behaviors and NPDs, we employed a genetically informed MR approach. Our results suggest that mentally passive sedentary behaviors such as television watching are risk factors for MDD, whereas mentally active sedentary behaviors such as computer use act as protective factors against AD. In the reverse direction, we found that SCZ reduces the time spent driving. The LDSC results were inconsistent with the MR analysis results, as the LDSC showed a significant positive relationship between television watching and MDD and a negative relationship between computer use and driving and AD and SCZ.

Our results build upon existing studies in several ways. First, we examined the effects of different sedentary behavior domains (mentally passive (i.e., television watching) and mentally active (i.e., computer use and driving) behaviors) on NPD risk and thereby discovered that mentally passive and mentally active sedentary behaviors have different relationships with NPDs. Specifically, we found that television watching, a mentally passive behavior, is causally associated with an increased MDD risk [7,17], and that computer use, a mentally active behavior, is causally associated with a decreased AD risk [16,38]. These findings are consistent with recent studies reporting that mentally active sedentary behaviors and mentally passive sedentary behaviors have different effects on NPDs [39,40]. In addition, a meta-analysis showed that television watching is positively associated with MDD risk, whereas computer use is not associated with MDD [7]. Moreover, a prospective study showed that computer-related sedentary behavior is positively related to cognitive function [16]. Similar results were obtained from meta-analyses, demonstrating that sedentary behaviors are significantly related to an increase in AD risk [38] and cognitive decline [8]. However, another meta-analysis found that there is no causal association between sedentary behaviors, physical activities, and SCZ [40]. These discrepant results highlight the importance of distinguishing mentally active sedentary behaviors from mentally passive sedentary behaviors to understand the causal relationships between sedentary behaviors and NPDs [16]. Thus, a lack of distinction between the different domains of sedentary behaviors may explain some of the disparities between studies.

Several mechanisms could lead to the relationship between sedentary behaviors and MDD and AD. First, mentally passive sedentary behaviors can hinder direct communication between individuals, reducing social interactions and increasing the risk of incidence of depression [41]. Second, sedentary behaviors reduce the time spent on physical exercise, which is demonstrated to effectively prevent and treat depression. Interestingly, we found that computer use, a mentally active behavior, could benefit cognitive function. This is consistent with the fact that computer use has been shown to enhance memory, concentration, and executive function [42,43].

Our study also aimed to elucidate the direction of the causal relationships between sedentary behaviors and NPDs. Using bidirectional MR, we found evidence that SCZ risk was causally related to reduced driving time. However, a previous study indicated that people with SCZ engaged in more sedentary behaviors [44,45]. This inconsistency may be attributed to recognizing driving as a mentally active behavior that requires focus and behavior control [45,46]. People with SCZ often have difficulty regulating their emotions and behaviors, which can mean they find it difficult to drive [47]. Moreover, reliance on self-reporting may cause inaccurate estimates of sedentary behaviors in patients with SCZ, which may be exacerbated by the cognitive impairment associated with the disorder [5].

### Limitations and Strengths

First, despite the fact the instrumental variables in our study were strongly related SNPs, common SNPs have not yet been shown to explain much of the total variance (from 0.04% to 1.70%; presented in Appendix A) in complex diseases and thus cannot be considered exact proxies for an exposure. Second, excluding entirely pleiotropic mechanisms is impossible because we do not fully understand the functional biological actions of these SNPs. Although pleiotropy in the horizontal direction is a problem for MR inference, pleiotropy in the vertical direction where an exposure acts on an outcome through other factors along an identical causal pathway is acceptable. Third, setting a strict *p*-value threshold for including SNPs that are strongly associated with exposure to meet the assumption 1 of MR would include fewer SNPs into the primary analysis, such as driving (only 4 SNPs left in the further study), which may lead to a decreased power to conclude. However, we performed the additional MR analyses with a tolerant *p*-value threshold for exposures, such as driving to include more SNPs into the primary analysis. We got the same conclusion: there was no causal relationship between driving and NPDs with sufficient power (>80%). 

Despite these limitations, our results from MR show that genetic instrumental variables could provide independent evidence for potentially risk-decreasing or risk-increasing relationships between sedentary behaviors and NPDs. Moreover, our results highlight the importance of separately assessing mentally passive and active sedentary behaviors to promote the understanding of the association between sedentary behaviors and NPDs. Furthermore, MR analysis adopting genetic variants as instruments for causal estimates removes the traditional challenges of observational studies and strengthens the power. More robust evidence for causal relationships between sedentary behaviors and NPDs is essential because most modifiable variables for preventing NPDs are unknown. Promoting mentally active sedentary behaviors as alternatives to mentally passive sedentary behaviors may serve as a specific NPD preventive strategy for healthy individuals and for those at risk of developing NPDs.

## 5. Conclusions

This study leveraged MR to explore causal inferences regarding putative protective or risk factors for NPDs, which helps circumvent residual confounding, measurement errors, and reverse causation. The results fit in with our observations and prior knowledge and support the wealth of epidemiological studies validating that television watching is causally associated with an increased risk of MDD, and that computer use is causally related to a decreased risk of AD. In the opposite direction, we detected a possible causal association between SCZ and reduced driving time. Our findings re-affirm these risk associations as well as highlighting the importance of distinguishing mentally passive from mentally active sedentary behaviors when exploring the causal relationships between sedentary behaviors and NPDs, which can inform the study design of interventions.

## Figures and Tables

**Figure 1 genes-13-00962-f001:**
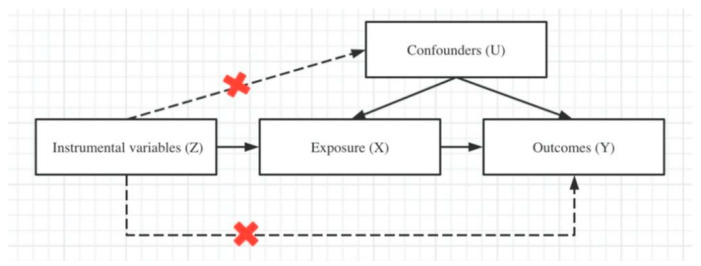
The framework of Mendelian randomization analysis and key assumptions.

**Figure 2 genes-13-00962-f002:**
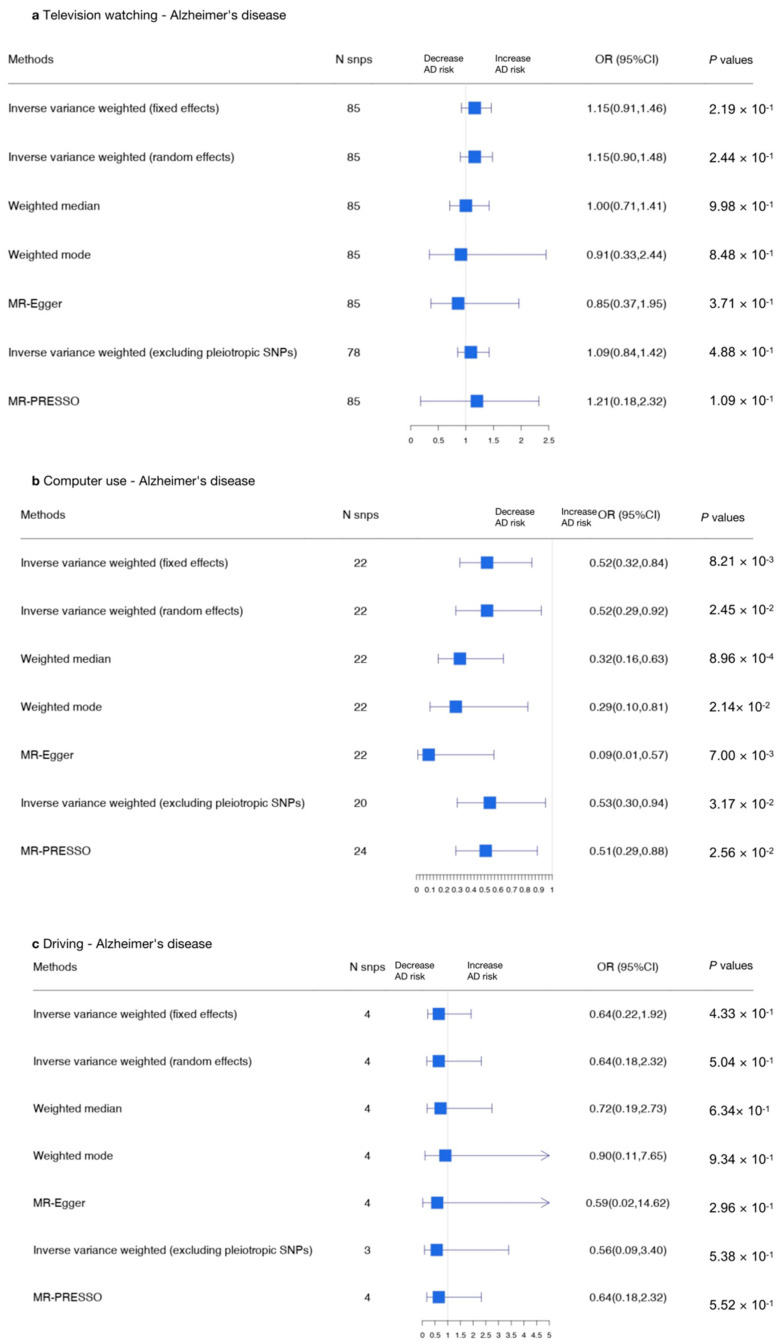
Summarized Mendelian randomization (MR) effect sizes between sedentary behaviors and AD. Summarized MR effect sizes of the causal relationship between (**a**) television watching time, (**b**) computer use time, and (**c**) driving time and AD were estimated using the following approach. The methods used in the analyses included IVW, IVW removing genetic variants with a potentially pleiotropic effect in any phenotype, MR-Egger, weighted median, MR pleiotropy residual sum and outlier (MR-PRESSO), outlier-corrected MR-PRESSO, and weighted mode methods. Odds ratios (ORs) as well as 95% confidence intervals (95% CIs) are presented on the x axis. A two-sided *p*-value of less than 0.05 was considered statistically significant.

**Figure 3 genes-13-00962-f003:**
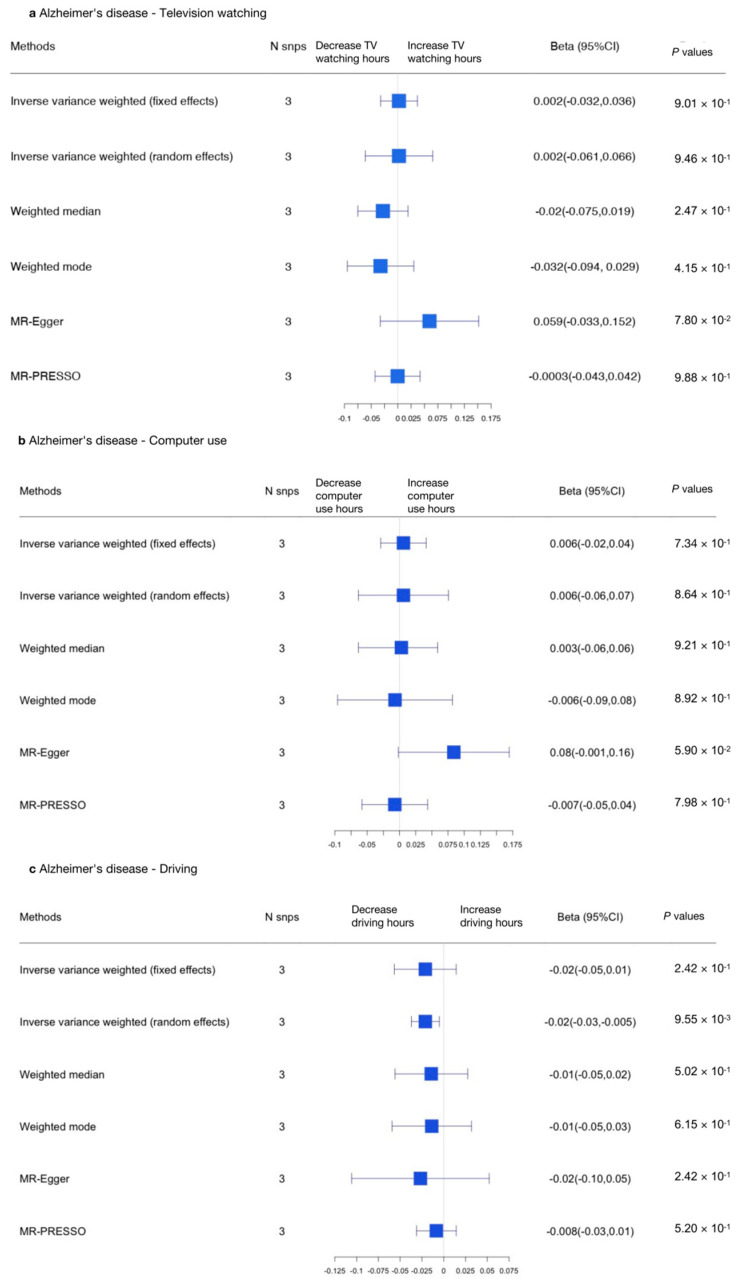
Summarized Mendelian randomization (MR) effect sizes of AD and sedentary behaviors. Summarized MR effect sizes of the causal relationship between AD and (**a**) television watching time, (**b**) computer use time, and (**c**) driving time were estimated using the following approach. The methods used in the analyses included IVW, IVW removing genetic variants with a potentially pleiotropic effect in any phenotype, MR-Egger, weighted median, MR pleiotropy residual sum and outlier (MR-PRESSO), outlier-corrected MR-PRESSO, and weighted mode methods. Odds ratios (ORs) as well as 95% confidence intervals (95% CIs) are presented on the x axis. A two-sided *p*-value of less than 0.05 was considered statistically significant.

**Figure 4 genes-13-00962-f004:**
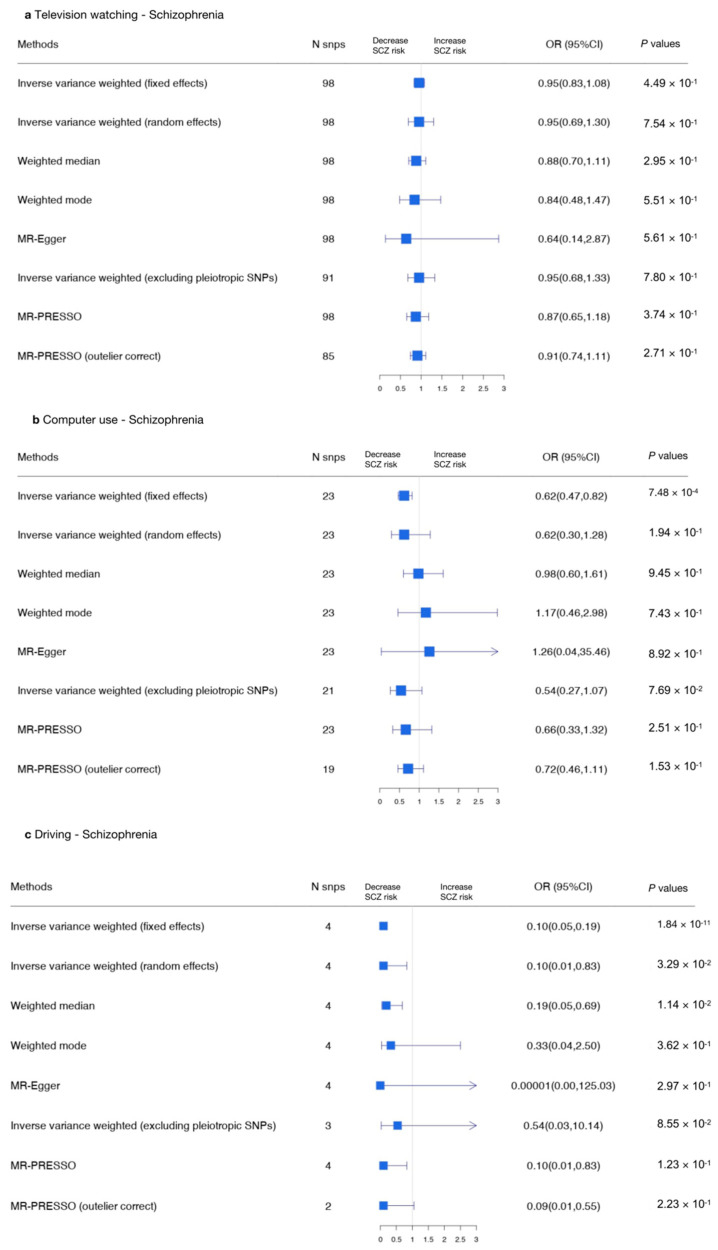
Summarized Mendelian randomization (MR) effect sizes between sedentary behaviors and SCZ. Summarized MR effect sizes of the causal relationship between (**a**) television watching time, (**b**) computer use time, and (**c**) driving time and SCZ were estimated using the following approach. The methods used in the analyses included IVW, IVW removing genetic variants with a potentially pleiotropic effect in any phenotype, MR-Egger, weighted median, MR pleiotropy residual sum and outlier (MR-PRESSO), outlier-corrected MR-PRESSO, and weighted mode methods. Odds ratios (ORs) as well as 95% confidence intervals (95% CIs) are presented on the x axis. A two-sided *p*-value of less than 0.05 was considered statistically significant.

**Figure 5 genes-13-00962-f005:**
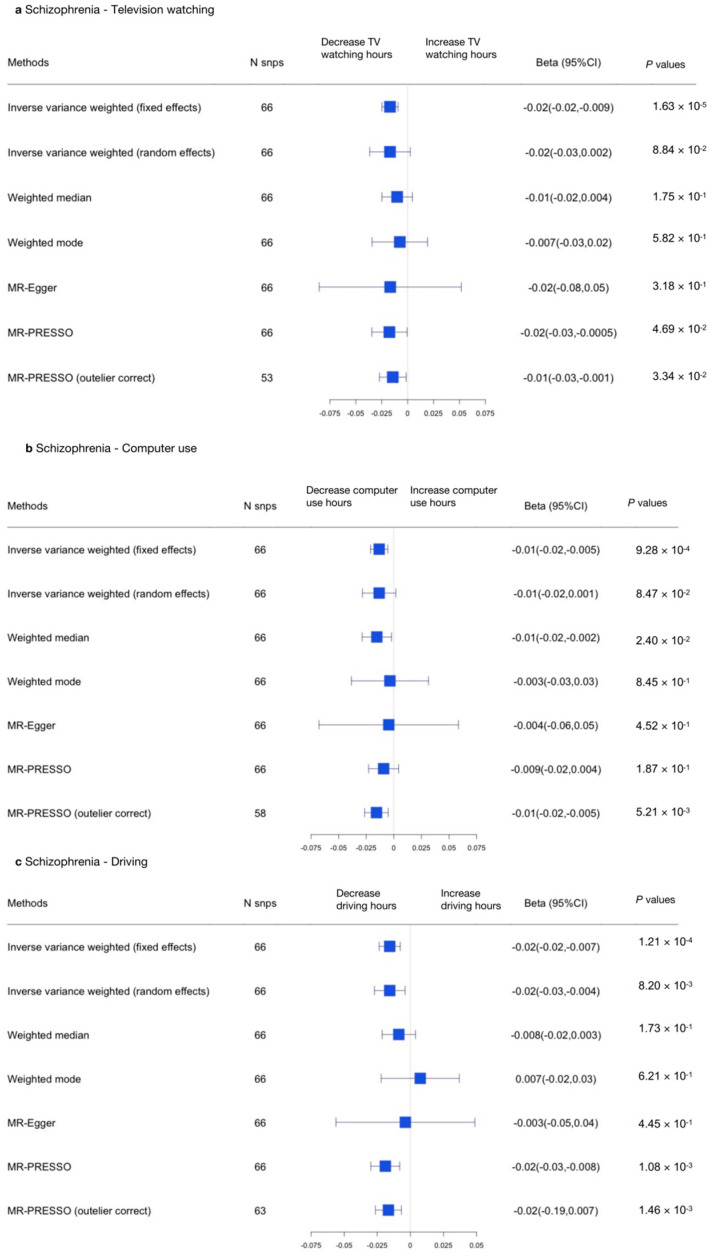
Summarized Mendelian randomization (MR) effect sizes of SCZ and sedentary behaviors. Summarized MR effect sizes of the causal relationship between SCZ and (**a**) television watching time, (**b**) computer use time, and (**c**) driving time were estimated using the following approach. The methods used in the analyses included IVW, IVW removing genetic variants with a potentially pleiotropic effect in any phenotype, MR-Egger, weighted median, MR pleiotropy residual sum and outlier (MR-PRESSO), outlier-corrected MR-PRESSO, and weighted mode methods. Odds ratios (ORs) as well as 95% confidence intervals (95% CIs) are presented on the x axis. A two-sided *p*-value of less than 0.05 was considered statistically significant.

**Figure 6 genes-13-00962-f006:**
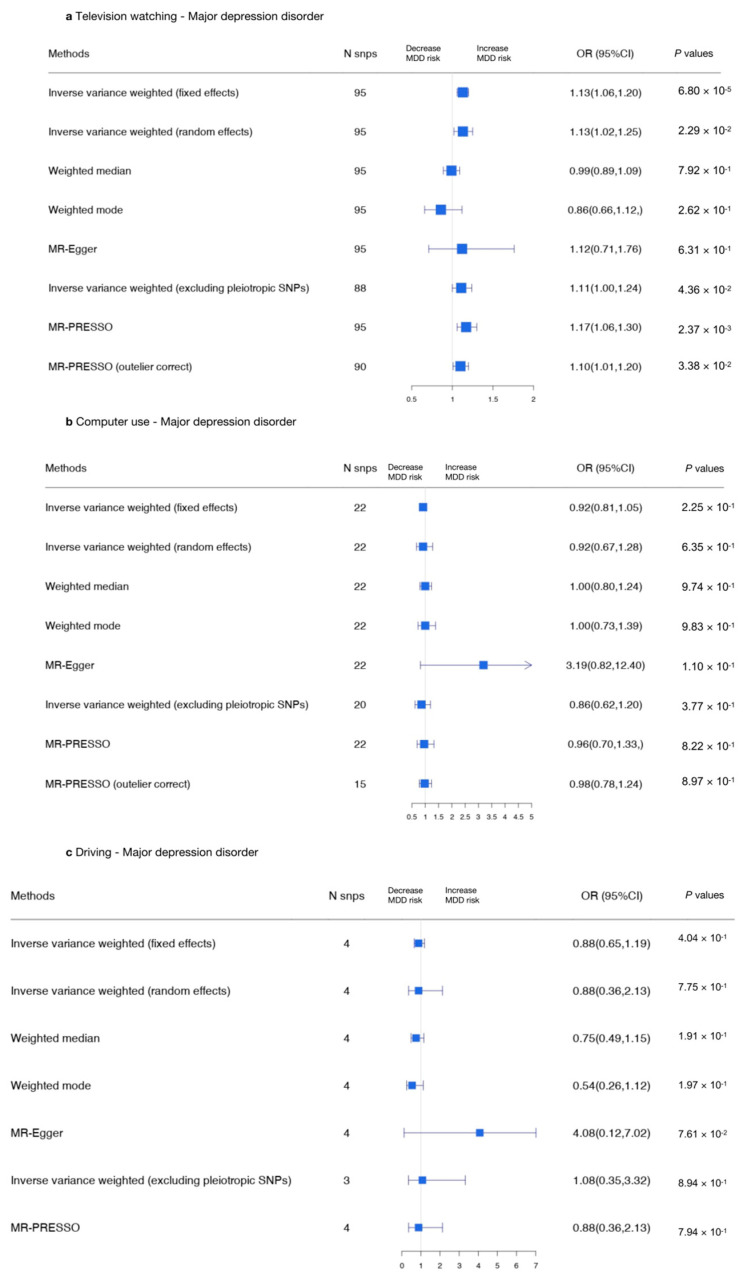
Summarized Mendelian randomization (MR) effect sizes between sedentary behaviors and MDD. Summarized MR effect sizes of the causal relationship between (**a**) television watching time, (**b**) computer use time, and (**c**) driving time and MDD were estimated using the following approach. The methods used in the analyses included IVW, IVW removing genetic variants with a potentially pleiotropic effect in any phenotype, MR-Egger, weighted median, MR pleiotropy residual sum and outlier (MR-PRESSO), outlier-corrected MR-PRESSO, and weighted mode methods. Odds ratios (ORs) as well as 95% confidence intervals (95% CIs) are presented on the x axis. A two-sided *p*-value of less than 0.05 was considered statistically significant.

**Figure 7 genes-13-00962-f007:**
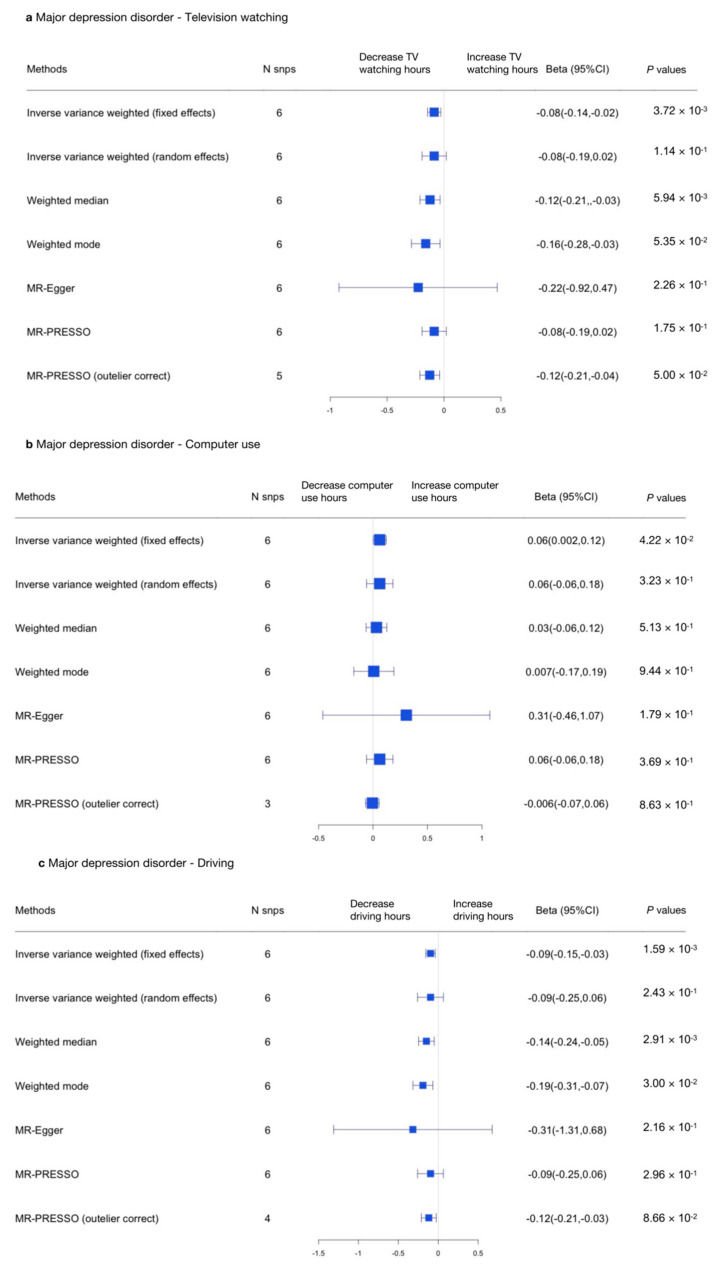
Summarized Mendelian randomization (MR) effect sizes of MDD and sedentary behaviors. Summarized MR effect sizes of the causal relationship between MDD and (**a**) television watching time, (**b**) computer use time, and (**c**) driving time were estimated using the following approach. The methods used in the analyses included IVW, IVW removing genetic variants with a potentially pleiotropic effect in any phenotype, MR-Egger, weighted median, MR pleiotropy residual sum and outlier (MR-PRESSO), outlier-corrected MR-PRESSO, and weighted mode methods. Odds ratios (ORs) as well as 95% confidence intervals (95% CIs) are presented on the x axis. A two-sided *p*-value of less than 0.05 was considered statistically significant.

**Table 1 genes-13-00962-t001:** Genetic correlation of sedentary behaviors and MPDs.

		Constrained Intercept			Unconstrained Intercept		
		Television Watching	Computer Use	Driving	Television Watching	Computer Use	Driving
Outcome: AD							
Cross-trait	Genetic correlation (*r_g_* ± SE)	0.1500 ± 0.0259	−0.1741 ± 0.0258	0.0926 ± 0.0338	0.2232 ± 0.0701	−0.2475 ± 0.0675	0.1112 ± 0.0815
LDSC	*Pr_g_*	6.74 × 10^−9^	1.59 × 10^−11^	6.08 × 10^−3^	1.50 × 10^−3^	2.00 × 10^−4^	1.72 × 10^−1^
Outcome: SCZ							
Cross-trait	Genetic correlation (*r_g_* ± SE)	−0.08 ± 0.0119	−0.0747 ± 0.0135	−0.1581 ± 0.0171	−0.0981 ± 0.0175	−0.0682 ± 0.0209	−0.1869 ± 0.0272
LDSC	*Pr_g_*	1.97 × 10^−11^	3.28 × 10^−8^	3.04 × 10^−20^	2.02 × 10^−8^	1.12 × 10^−3^	5.93 × 10^−12^
Outcome: MDD							
Cross-trait	Genetic correlation (*r_g_* ± SE)	0.194 ± 0.0128	0.0392 ± 0.0169	−0.1611 ± 0.0200	0.125 ± 0.0193	0.0182 ± 0.0257	−0.0652 ± 0.0285
LDSC	*Pr_g_*	9.87 × 10^−52^	2.03 × 10^−2^	1.02 × 10^−17^	9.24 × 10^−11^	4.77 × 10^−1^	2.22 × 10^−2^

## Data Availability

The datasets of leisure sedentary behaviors for this study can be found in the Mendeley data (https://data.mendeley.com/datasets/mxjj6czsrd/1, accessed on 10 September 2021). The dataset of AD can be found in the IGAP consortium (https://consortiapedia.fastercures.org/consortia/igap/, accessed on 10 September 2021). The datasets of SCZ and MDD can be found in the PGC (https://www.med.unc.edu/pgc/, accessed on 15 September 2021).

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
