# Peer review of "Assessment of Bidirectional Relationships between Leisure Sedentary Behaviors and Neuropsychiatric Disorders: A Two-Sample Mendelian Randomization Study"

_genes, 2022, doi:10.3390/genes13060962_

Round 1

Reviewer 1 Report

This study by He et al uses two-sample Mendelian randomization (MR) to identify causal relationships between sedentary behaviors and neuropsychiatric disorders, namely Alzheimer’s disease (AD), schizophrenia (SCZ), and major depressive disorder (MDD). The authors find causal relationships between computer use and decreased AD risk, between TV watching and increased MDD risk, and between SCZ and decreased riving time. The paper is very well written, clearly illustrated, and easy to follow.

Major issues:

  1. For driving phenotype, only 4 SNPs were used for MR. Will the small number affect the rigor of the analysis? 

  1. Confusing statements:
  1. In line 237, it was stated that “no evidence of causal relationships between AD and sedentary behaviors”, while there is a significant causal relationship between computer use and decreased AD risk.;
  2. Similar to a, in line 351, it was stated that “no evidence of causal relationships between MDD and sedentary behaviors”, while there is a significant causal relationship between TV watching and increased MDD risk;
  3. For better clarity, I suggest that the authors highlight the significant causality findings in the first paragraph of each of the three subsections of section 3.2.

  1. In figure 3, panel a and c’s p values are identical. Please double-check and correct. Correspondingly, p = 0.901 in line 286 is not shown in the figure.

Minor issues:

  1. Typos: 
  1. Add columns after the four sections in the abstract (lines 17, 21, 28, and 34);
  2. Remove 21 in line 114;
  3. Remove 34 in lines 191 and 192;
  4. Add a full stop in line 441 between depression and Interestingly.

  1. Choose a better resolution for the main figures.

Author Response

Dear Reviewer:

Thank you very much for your helpful comments on this manuscript. We have addressed most if not all of the comments (please see the point-by-point response to the reviewers_Genes.pdf] raised by reviewers and resubmitted the manuscript for possible publication in Genes; if you have any questions about the present work, please do not hesitate to contact us.

Reviewer 2 Report

Comments on “Assessment of bidirectional relationships between leisure sedentary behaviors and neuropsychiatric disorders: A two-sample Mendelian randomization study (genes-1721128)”

He et.al. used a two-sample MR analysis strategy examined the causality and direction between leisure sedentary behaviors and neuropsychiatric disorders, such as Alzheimer's disease, schizophrenia, and major depressive disorders. The results demonstrated that television watching causally associated with risk of MDD while computer use causally associated with decreased risk of AD. In contrast, SCZ may be causally associated with reduced driving time. This seems a well-conducted study, the results were presented clearly, and the manuscript is well-written.

Author Response

Thank you very much for your helpful comments on this manuscript. We have addressed most if not all of the comments [Please see the point-by-point response to reviewers_Genes.2.pdf] raised by reviewers and resubmitted the manuscript for possible publication in Genes; if you have any questions about the present work, please do not hesitate to contact us.
